# Development of Magnetic-Based Navigation by Constructing Maps Using Machine Learning for Autonomous Mobile Robots in Real Environments

**DOI:** 10.3390/s21123972

**Published:** 2021-06-09

**Authors:** Takumi Takebayashi, Renato Miyagusuku, Koichi Ozaki

**Affiliations:** Graduate School of Engineering, Utsunomiya University, 7-1-2 Yoto, Utsunomiya, Tochigi 321-8585, Japan

**Keywords:** machine learning, Gaussian processes, autonomous mobile robots, robust navigation, mapping, localization

## Abstract

Localization is fundamental to enable the use of autonomous mobile robots. In this work, we use magnetic-based localization. As Earth’s geomagnetic field is stable in time and is not affected by nonmagnetic materials, such as a large number of people in the robot’s surroundings, magnetic-based localization is ideal for service robotics in supermarkets, hotels, etc. A common approach for magnetic-based localization is to first create a magnetic map of the environment where the robot will be deployed. For this, magnetic samples acquired a priori are used. To generate this map, the collected data is interpolated by training a Gaussian Process Regression model. Gaussian processes are nonparametric, data-drive models, where the most important design choice is the selection of an adequate kernel function. These models are flexible and generate mean predictions as well as the confidence of those predictions, making them ideal for their use in probabilistic approaches. However, their computational and memory cost scales poorly when large datasets are used for training, making their use in large-scale environments challenging. The purpose of this study is to: (i) enable magnetic-based localization on large-scale environments by using a sparse representation of Gaussian processes, (ii) test the effect of several kernel functions on robot localization, and (iii) evaluate the accuracy of the approach experimentally on different large-scale environments.

## 1. Introduction

The new coronavirus, which caused a pandemic that raged around the world in 2020, has made us change the way we have always lived. It is recommended that face-to-face conversations and meetings are avoided, and travel by public transportation now poses a risk of viral infection. Common practice has changed, and technological innovations have become key to enable us to cope with this new reality. In particular, robotic research has enabled robots to be used in a wide variety of applications, such as delivery and customer service [1,2]. It is expected that in the near future autonomous mobile robots will replace human labor, providing services such as transportation and security in indoor public spaces. This lowers the risks of contagion of workers, e.g., security guards, by lowering the chances of contact with customers in commercial facilities [3,4] and can make it possible to use autonomous mobile robots to supplement the labor force after COVID-19, making the development of real applications using autonomous mobile robots appealing due to the plethora of benefits it can bring to society.

To perform navigation, it is necessary to develop technologies such as mapping, localization, and path planning [5]. In this work, we focus on accurate and reliable localization in large-scale real environments for stable navigation.

Several sensors can be used for robot localization, such as Global Positioning System (GPS) [6,7], 2D and 3D laser rangefinders [8,9], monocular and RGB-D cameras [10,11], wireless signals [12,13], and magnetic sensors [14]. For indoor localization in particular, a notable exception is the use of GPS, as radio waves from satellites are often blocked, making localization impossible [15].

Laser rangefinders and cameras compare acquired information (range or features extracted from images) to a previously built map of the environment. Approaches using these sensors are typically very accurate; however, in crowded indoor public spaces such as supermarkets, people become dynamic obstacles that constantly occlude sensor readings, making localization accuracy deteriorate [16,17].

Wireless signals and magnetic data can also be used for robot localization by matching observed sensor data to a previously made strength map. These techniques are collectively referred to as fingerprinting, scene analysis, or profiling techniques. In the case of wireless signal strength, this is typically a map of received signal strength information (RSSI) [13] or channel state information (CSI) [18]; while for magnetic data it is magnetic strength in a given axis [19] or magnetic angle [20].

For fingerprinting, sensor samples are first collected at known locations of the environment. These locations can be generated using other sensors, such as laser rangefinders [21] when there are no or few people in the environment, or map generation and localization can be done simultaneously by (Simultaneous Localization and Mapping (SLAM) [22,23]. These data points are collected onto a training dataset that is used to predict the location of new measurements by either matching new observed sensor data to the most similar samples on the training dataset, becoming a classification problem, or by generalizing these points by learning continuos or grid-map of the environment, making it a regression problem.

Previous works using classification techniques include the usage of *k*-Nearest Neighbors [24,25], support vector machines [26,27], random forests [28], among others. While works using regression techniques include linear interpolation in graphs [29], smoothing functions [30], Gaussian Processes Regression (GPR) [12,31], among others.

Wireless signal strength, especially on the 2.4 GHz bandwidth, does not allow for precise localization (even considering multiple sources and proper data fusion, accuracy is around 1 m on indoor environments [32]), mainly due to wireless signal strength distribution being wide (strength does not change fast with respect to the location signals that were taken). Therefore, WiFi-only robot navigation would lead to accidents such as collisions with obstacles when the robot moves autonomously for services such as delivery and customer service. Instead, it is typically used with other sensors [20,21,33], where WiFi is used for enhanced robustness. Additionally, while wireless signals are not completely occluded by obstacles (even walls and furniture), signal strength does lower, especially in the presence of crowds of people, making the previously acquired map information less accurate [34].

The magnetic information we refer to in this work is the strength and direction of the environmental magnetic field, which is a combination of geomagnetism and residual magnetism. Geomagnetism is the Earth’s magnetic field and generally has a constant intensity and always points towards the magnetic north. Residual magnetism refers to the magnetic disturbance caused by magnetic materials in the environment, such as steel frames, pipes, and manholes.

These materials often remain magnetized after being exposed to magnetic fields during their manufacturing or construction processes of the building where they are placed. Due to the effect of this residual magnetism, the environmental magnetic field has several local disturbances in strength and orientation. Some robots using electronic compasses model these disturbances as noise taking measurements to avoid them. However, as these magnetic disturbances are stable over time [35], they can be used as landmarks for localization. It is important to notice that, if new magnetic materials are fixed in the environment, the distribution of the magnetic intensity will change. In such a case, it is necessary to acquire new magnetic intensity measures in the vicinity of the area, to update the training points that changed, and to learn the mapping again. This can be done before the next navigation task or simultaneously by using SLAM. In this paper, we do not discuss the solutions to these problems.

These landmarks can be used to correct odometry information [36,37] or for localization using fingerprinting techniques.

Magnetic fingerprinting first obtains magnetic samples at known locations of the environment. Using these training samples, a magnetic map is learned. Then, similarly to laser rangefinders or cameras, localization is performed by comparing new magnetic measurements to this learned map. The locations of training samples can be generated using other sensors, such as laser rangefinders when there are no or few people in the environment, or map generation and localization can be done simultaneously (Simultaneous Localization and Mapping; SLAM) [23].

As magnetic changes are more spatially localized than WiFi signal changes, higher localization accuracies are possible. Furthermore, as magnetic information is not affected by nonmagnetic materials, it is reliable even in crowded environments [14]. This makes magnetic fingerprinting more suitable for navigation in crowded environments than any other of the aforementioned sensors. In this work, we employ Gaussian Processes Regression (GPR) for magnetic-based localization. GPR is an approach that has been successfully used to learn magnetic maps [19,38,39]. GPR is a data-driven, nonparametric model, that is fully defined by a mean and a kernel function. Mean functions for GPR are often set to the zero function, especially when modeling disturbances [40]. Therefore, the single most important design choice for magnetic-based localization using GPR is the adequate selection of a kernel function. Therefore, in this research, we explore the effect of different kernel functions on magnetic map generation and how this affects localization accuracy.

While GPR is a flexible model estimation approach, it scales poorly (with respect to the amount of data used for training), requiring large amounts of memory when training. For environments like the ones available on the UJIIndoorLoc-Mag dataset [41], where the number of acquired data points on each environment is relatively small, learning can be done using GPR without requiring prohibitive amounts of memory (Table 1 shows learning time and memory consumption for the UJIIndoorLoc-Mag dataset). However, if the environment to be magnetically mapped is extensive and the path taken to acquire the magnetic data is long, the number of acquired magnetic data becomes too large. In such a case, the required amount of memory for learning a GPR model becomes prohibitive even for modern servers. To alleviate this problem, some WiFi-localization approaches have proposed filtering data points and eliminating redundant access points [42] as well as learning compact representations (to reduce the required number of dimensions necessary to represent all the access points in the environment) [43]. Unfortunately, these approaches cannot be applied for magnetic localization, as the problem for large-scale magnetic mapping is not data dimensionality (1 magnetic axis in our case, opposed to hundreds of access points for WiFi-localization) but rather the necessary number of data points required for accurate modeling of sharp strength distributions.

In this study, we propose the use of Sparse Gaussian Processes Regression (SGPR) to reduce memory costs and realize magnetic mapping of a large-scale environment such as a commercial facility.

To the extent of our knowledge, this is the first work that addresses the generation of magnetic maps for large-scale environments using a GPR-based approach as well as the relation between kernel functions and robot accuracy in such a case.

The remainder of this paper is organized as follows: Section 2 introduces our approach for robot localization using magnetic disturbances. Section 4 describes the Gaussian and sparse Gaussian processes approach used for magnetic mapping. Section 5 describes the experiments on the localization accuracy by selecting the kernel function and the magnetic mapping of a large-scale environment using the sparse Gaussian process. Section 6 gives our conclusions.

## 2. Magnetic Navigation Method

In this work, we address robot localization in a x−y Cartesian coordinate. The robot localization problem consists of determining a robot’s pose ([x,y,θ]) relative to a given map of the environment. Among the several approaches that have been introduced in the literature, we adopt the Monte Carlo Localization (MCL) algorithm [44]. MCL is an implementation of the Bayes Filter, using a particle filter, which uses *particles* to represent possible robot poses, which are distributed according to the probability density estimation of the robot’s pose conditioned on a time-series data of believed previous poses, robot movements, and sensor measurements. In our case, these sensor measurements are magnetic information corresponding to the robot’s *z*-axis, *m*, as proposed in [19]. The magnetic strengths on the *x* and *y*-axis are not used because the observed values change depending on the robot posture. Figure 1 shows a schematic diagram of localization by the magnetic localization.

Magnetic localization using MCL is performed iteratively by conditioning robot poses’ belief to magnetic measurements by comparing acquired sensor data to predicted values from a previously generated magnetic map, to then updating the belief of the robot’s location:Updating the location particles given the robot’s movement.Computing the likelihood of each particle based on the most current sensor measurement and the observation model
(1)$$\omega =\frac{1}{\sqrt{2\pi \sigma }}\exp\left(-\frac{{\left(z_{map}-z_{sensor}\right)}^{2}}{2\sigma }\right)$$
where ω is particle’s likelihood, σ is the variance of the observations, $z_{map}$ is the magnetic information on the map, $z_{sensor}$ is observation acquired by magnetic sensor.Resampling particles according to ω when necessary.

Particles are not always resampled as it has been reported that resampling is effective for improving performance if executed only when the likelihood bias in the particle group is large [45]. Specifically, we resample particles when the Effective Sample Size (ESS) of the particles is lower than a threshold [46].

ESS is an index that estimates the number of effective particles contained in the particles that can be used as an estimator of the bias in the likelihood of particles and is computed as
(2)$$n_{ess}=\frac{1}{\sum_{i=1}^{n}\omega_{i}^{2}}.$$

## 3. Machine Learning for Generating Magnetic Map

When using a robot to survey an area, the magnetic information is not captured as contiguous data but as discrete data points by the magnetic sensor. Therefore, regions with unmeasured magnetic information exist, even if we move through the entire environment. If robots use the captured magnetic information as the map, they would not be able to perform localization because of these regions. Hence, we need to develop a way of generating dense magnetic maps without unmeasured regions by interpolating magnetic information.

### 3.1. Gaussian Process Regression (GPR)

A magnetic map can be learned using training data acquired directly from the desired environment using Gaussian Process Regression (GPR). GPR is a generalization of normal distributions to functions, describing functions of finite-dimensional random variables. It is a data-driven approach that given training points, learns the correlation between them using a kernel function and generalizes a continuous distribution from them [47].

In our case, we define (X,m) as the training data, where $m\in R^{n\times 1}$ is a vector of *n* magnetic samples *m* (only the *z*-axis component of the magnetic field) acquired in the environment, and $X\in R^{n\times 2}$ is the matrix of the corresponding x−y locations, where the samples were obtained.

Under the GPR formulation, each data pair $\left(x_{i},m_{i}\right)$ is assumed to be drawn from a process with i.i.d Gaussian noise, and any two output values, $m_{p}$ and $m_{q}$, are assumed to be correlated by a covariance function based on their input values $x_{p}$ and $x_{q}$, so that
(3)$$cov\left(m_{p},m_{q}\right)=k\left(x_{p},x_{q}\right)+\sigma_{n}^{2}\delta_{pq}$$

$k(x_{p},x_{q})$ is a kernel function, $\sigma_{n}^{2}$ is the variance of the i.i.d Gaussian noise, and $\delta_{pq}$ is one only if p=q and zero otherwise.

Then, the expected values of unknown data point $x_{*}$, conditioned on training data (X,m) can be estimated as,
(4)$$p_{\mathrm{GPR}}\left(m_{*}|x_{*},X,m\right)\;∼\;N\left(E_{\mathrm{GPR}}\left[m_{*}\right],{\mathrm{var}}_{\mathrm{GPR}}\left(m_{*}\right)\right),$$
where,
(5)$$\begin{array}{c} E_{\mathrm{GPR}}\left[m_{*}\right]=k_{*}^{T}{\left(K+\sigma_{n}^{2}I_{n}\right)}^{-1}m, \end{array}$$
(6)$$\begin{array}{c} {\mathrm{var}}_{\mathrm{GPR}}\left(m_{*}\right)=k_{**}-k_{*}^{T}{\left(K+\sigma_{n}^{2}I_{n}\right)}^{-1}k_{*}, \end{array}$$
where K=cov(X,X) is the n×n covariance matrix between all training points X; $k_{*}$ = $cov(X,x_{*})$ is the covariance vector that relates the training points X and the test point $x_{*}$; $k_{**}$ = $cov(x_{*},x_{*})$ is the variance of the test point; and $I_{n}$, is the identity matrix of rank *n*. Therefore, predictions from a GPR are fully defined by the acquired training data and the selected kernel function. Since there are various types of kernel functions, kernel selection is the single most important design choice given available training data.

### 3.2. Sparse Gaussian Process Regression (SGPR)

The previously described GPR formulation can flexibly predict the function for a given input using Equation (Equation 5). On a closer inspection of this equation, it can be observed that the expected value can be obtained by performing the following procedure on the input data:Compute the vector $k_{*}$ and matrix $K+\sigma_{n}^{2}I_{n}$ and store them in memory.Compute the inverse matrix ${\left(K+\sigma_{n}^{2}I_{n}\right)}^{-1}$ and store it in memory.Calculate $k_{*}^{T}{\left(K+\sigma_{n}^{2}I_{n}\right)}^{-1}m$ and store it in memory.

It is known that the inverse matrix calculation in Procedure 2 has the highest memory consumption and the highest order of arithmetic operations in performing these calculations, $O(N^{3})$, which is of the order of arithmetic operations, if the calculation takes 1 s for N=1k, it takes 8 s for N=2k, and 1000 s for N=10k, which is about 17 min. This bottleneck has been cited as a drawback and impracticality of GPR for addressing problems with a large number of training data. To solve this problem, various methods have been proposed to reduce the number of required computations and improve efficiency [48].

In this study, we use the inducing variable method, which is one of the major methods previously proposed to address the aforementioned GPR drawbacks. In the inducing variable method, *M* (M<N) inducing variables are introduced to approximate the GPR covariance matrix, compressing it to M×M matrices. By doing so, we can reduce the order of memory consumption to $O(NM+M^{2})$ and the order of computation to $O(NM^{2}+M^{3})$.

#### 3.2.1. Subset of Data Approximation (SoD)

A different approach is the partial data method. This partial data method is the basis of the inducing variable method described above. From the *N* input data points, *M* input points (M<N) are selected to represent the distribution of all data well, and only these *M* input points are used for training. Therefore, the inverse matrix calculation of N×N originally required can be reduced to M×M to obtain the predictive distribution of the Gaussian process regression. In situations where the selected sub-data consisting of *M* input data points are well representative of the total data, the same level of accuracy can be obtained at a much lower computational cost $O(M^{3})$ than $O(N^{3})$.

#### 3.2.2. The Inducing Variables Method

In the inducing variable method, *M* (M<N) virtual input points called inducing input points $Z=(z_{1},\ldots ,z_{M})$ are appropriately placed within the definition of the function f(·).

This partial data is then used to efficiently estimate the function values at the test data points. SoD does not use the other (N−M) data points but can maintain accuracy as if using all the data.

The output value $f(z_{m})$ at inducing point $z_{m}$ is called inducing variable $u_{m}=f\left(z_{m}\right)$, which is summarized in a vertical vector u with respect to m=1,…,M. By using inducing input points, the predictive distribution of the unknown function f(x) can be obtained. There are various approaches to obtain inducing variables; in this study, we adopt the Fully Independent Training Conditional (FITC) method [49]. Unlike other approaches, FITC is a method that predicts the distribution by also considering the calculation of the variance. Since the magnetic navigation method also calculates the likelihood using the variance of the predicted distribution in the Gaussian process, a method that approximates the calculation of the variance may affect the localization accuracy.

The Gaussian process described in Section 3.1 uses the following covariance matrix for inverse matrix calculation, etc.
(7)$$K_{\mathrm{GPR}}=\left(\begin{array}{cc} K_{NN} & k_{N*}\\ k_{N*}^{T} & k_{**} \end{array}\right)$$

In contrast, the covariance matrix in FITC takes the following form
(8)$$K_{\mathrm{FITC}}=\left(\begin{array}{cccc} k_{1} & \; & O & \\ \; & \ddots & \; & k_{NM}\\ O & \; & k_{n} & \\ \; & k_{NM}^{T} & \; & k_{MM} \end{array}\right)$$

Comparing this covariance matrix with Equation (Equation 7), the (N×N) covariance matrix of the prior distribution f in the original Gaussian process, $K_{NN}$, is replaced by zero in Equation (Equation 8), ignoring all of its nondiagonal components, and the diagonal components $k_{n}=k\left(x_{n},x_{n}\right)\;\left(n=1,\ldots ,N\right)$ are left. By ignoring all the N(N−1) nondiagonal components in this way, the computational complexity can be greatly reduced when *N* is large.

As a result, FITC is able to predict the distribution for the unknown data point $x_{*}$ by the following equation.
(9)$$p_{\mathrm{FITC}}\left(m_{*}|x_{*},X,m\right)=\int p\left(f_{*}|u\right)p\left(u|y\right)du=N\left(E_{\mathrm{FITC}}\left[m_{*}\right],{\mathrm{var}}_{\mathrm{FITC}}\left(m_{*}\right)\right)$$
where ${\hat{k}}_{*}=K_{*u}$, $\hat{K}=K_{uu}$ and
(10)$$\begin{array}{ccc} E_{\mathrm{FITC}}\left[m_{*}\right] & = & {\hat{k}}_{*}^{T}\Sigma k_{M}^{T}\Lambda^{-1}m, \end{array}$$
(11)$$\begin{array}{ccc} {\mathrm{var}}_{\mathrm{FITC}}\left(m_{*}\right) & = & k_{**}-Q_{**}+{\hat{k}}_{*}^{T}\Sigma {\hat{k}}_{*}, \end{array}$$
(12)$$\begin{array}{ccc} \Sigma & = & {\left(\hat{K}+k_{M}^{T}\Lambda^{-1}k_{M}\right)}^{-1}, \end{array}$$
(13)$$\begin{array}{ccc} \Lambda & = & diag(K-Q+\sigma_{n}^{2}I_{n}), \end{array}$$
(14)$$\begin{array}{ccc} Q & = & {\hat{k}}_{*}^{T}{\hat{K}}^{-1}{\hat{k}}_{*}. \end{array}$$

As described above, SGPR using the inducing variable method, which is a computational modification of GPR, can achieve significant reductions in memory consumption and computational complexity, as shown in Table 3.

## 4. Kernel Function for High-Accuracy Localization

### 4.1. Kernel Function

Kernel functions are key components in GPR that affect the interpolation result of magnetic maps [50]. In this work, we analyze seven different kernel functions and their effect on magnetic maps and magnetic-based localization on GPR and SGPR.

For this analysis the following kernel functions were chosen:(1)Radial Basis Function (RBF) Kernel
(15)$$k\left(x_{p},x_{q}\right)=\sigma_{f}^{2}\exp\left(-\frac{r}{2l^{2}}\right)$$(2)Exponential Kernel
(16)$$k\left(x_{p},x_{q}\right)=\sigma_{f}^{2}\left(1+\sqrt{3}r\right)\exp\left(-\sqrt{3}r\right)$$(3)Matern 3/2 Kernel
(17)$$k\left(x_{p},x_{q}\right)=\sigma_{f}^{2}\left(1+\frac{\sqrt{3}r}{l}\right)\exp\left(-\frac{\sqrt{3}r}{l}\right)$$(4)Matern 5/2 Kernel
(18)$$k\left(x_{p},x_{q}\right)=\sigma_{f}^{2}\left(1+\frac{\sqrt{5}r}{l}+\frac{5r^{2}}{3l^{2}}\right)\exp\left(-\frac{\sqrt{5}r}{l}\right)$$(5)Exponential + Cosine Kernel
(19)$$k\left(x_{p},x_{q}\right)=\sigma_{f}^{2}\left(1+\sqrt{3}r\right)\exp\left(-\sqrt{3}r\right)+\frac{\pi }{4}\cos\left(\frac{\pi r}{2}\right)$$(6)Exponential + Matern 3/2 Kernel
(20)$$k\left(x_{p},x_{q}\right)=\sigma_{f}^{2}\left(1+\sqrt{3}r\right)\exp\left(-\sqrt{3}r\right)+\sigma_{f}^{2}\left(1+\frac{\sqrt{3}r}{l}\right)\exp\left(-\frac{\sqrt{3}r}{l}\right)$$(7)RBF + Exponential + Matern 3/2 Kernel
(21)$$k\left(x_{p},x_{q}\right)=\sigma_{f}^{2}\exp\left(-\frac{r^{2}}{2l^{2}}\right)+\sigma_{f}^{2}\left(1+\sqrt{3}r\right)\exp\left(-\sqrt{3}r\right)+\sigma_{f}^{2}\left(1+\frac{\sqrt{3}r}{l}\right)\exp\left(-\frac{\sqrt{3}r}{l}\right)$$
where r expresses a distance between two vectors ($x_{p},x_{q}$), $r=|x_{p}-x_{q}|$.

Figure 2 shows an example of the interpolation results of Gaussian process regression using each kernel function.

The Radial Basis Function (RBF) kernel is a kernel function commonly used for machine learning in GPR and support vector machines; while kernel functions (2) to (4) are less commonly used, we wanted to thoroughly verify their effects on magnetic mapping. In addition, as the linear combination of kernel functions also yields valid kernel functions, we considered some linear combinations of kernels (kernels (5) to (7)).

### 4.2. Experiments and Discussions

To verify the effect of the different maps generated in the previous section, we conducted an experiment using the robot shown in Figure 3 [51,52]. This robot has two independent wheel drives with an outer shape of 70 × 68 × 119 cm${}^{3}$ and 98 kg of weight.

Odometry information was collected from each wheel-encoder and magnetic information was collected using a 3-axis magnetic sensor (3DM-DH) shown in Figure 3. However, for localization purposes, only the data collected from its *z*-axis was employed. Sensor specifications are detailed in Figure 3.

The magnetic sensor was mounted 34 cm forward from the center of the robot’s drive shaft with its *z*-axis in the ground direction. Sensor placement was selected to avoid magnetic disturbances from other electronic devices mounted on the robot (the sensor was placed at least 15 cm away from any other device), as these would adversely affect the quality of the magnetic information acquired.

To obtain accurate pose information for our training datasets, we also installed a laser rangefinder (UXM-30LXH-EWA) in front of the robot. It is important to note that the laser rangefinder was only used while obtaining training data and not while using magnetic-based localization.

We conducted our experiments on the second floor of Utsunomiya University’s *Robotics, Engineering, and Agricultural-technology Laboratory (REAL)*. Figure 4 shows this environment.

Magnetic information was acquired by making round trips around the experimental environment while pushing the robot. Sensor information: odometry, range data, and magnetic information were stored in the log file using the Robotics Operating System (ROS). Pose locations for the training datasets were generated using the front-mounted laser rangefinder and graph-based SLAM using Google’s cartographer [53]. Using this training dataset, a magnetic map was created for each of the seven kernel functions discussed in Section 4.1 using Python’s GPy library (SheffieldML/GPy: Gaussian processes framework in Python. https://github.com/SheffieldML/GPy (accessed on 1 March 2021)).

REAL: 5, 10, 20, 30, 40, 50HICity: 10, 50, 100, 500, 1K, 2K

#### 4.2.1. Magnetic Maps

Figure 5 shows the magnetic maps generated by each kernel function. In these figures, the color bar shows the strength of the magnetic intensity, and the black dots show the magnetic strength training data points.

As the most commonly used kernel is the RBF kernel, we use it to assess the other kernel functions.

Exponential kernelThe Exponential kernel generates magnetic maps that are considerably smoother than those generated by the RBF Kernel. Smoother maps have been shown to improve localization accuracy when combined with MCL [34]. However, the maps generated by this kernel seem to be too smooth, which could limit the ability of the localization algorithm to differentiate between nearby points (as both have similar intensities), lowering its potential localization accuracy.Matern 3/2 kernelContrary to the exponential kernel, the Matern 3/2 generates maps that accentuate magnetic disturbances (higher peaks and valleys), while still generating smooth maps. As it can be observed, compared to the RBF kernel, several peaks are combined into larger ones.Matern 5/2 kernelSimilar to the Matern 3/2 kernel, the Matern 5/2 kernel also generates maps that accentuate magnetic disturbances. However, it does not tend to combine peaks, showing the same patterns as the RBF Kernel. Compared with the RBF kernel and Matern 3/2 kernel, it is hard to assess which would yield higher localization accuracies, hence the requirement to test the actual localization accuracy that can be achieved with them.Exponential + Cosine kernelThe Cosine kernel has periodicity in one dimension, but when combined with other kernels, the result does not show such periodicity. The main idea when testing the Exponential + Cosine kernel was to see if the training found some periodicity in the data. As can be seen, when compared with the Exponential kernel, this kernel has no significant differences. This means that no such periodicities were dominant in the data.Exponential + Matern 3/2 kernelAs both the Exponential and the Matern 3/2 kernels showed similar maps, we combined them to see if their combination would increase localization accuracy. As expected, the resulting maps are smooth and somewhat in the middle between the exponential and Matern in terms of the height of its peaks and valleys.Exponential + Cosine + RBF kernelThe RBF, Exponential, and Cosine kernels were combined to see if the addition of several kernels would improve localization accuracy.

#### 4.2.2. Localization Accuracy Using Different Magnetic Maps

To compare the localization accuracy when using different kernel functions, an experiment dataset was collected at a later date. Using the same robot and ROS, experiment data was logged and the ground truth locations of the robot were computed using Google’s cartographer.

Experiments were performed by replaying the logged data in real-time, and localization accuracy was computed using the magnetic localization system explained in Section 2.

Localization accuracy was computed as the root mean squared error (RMSE) between localization predictions by the MCL and the recorded ground truths. As an example, Figure 6 shows the result of the localization experiment obtained with the RBF kernel. In the same way, experiments are performed for the other six kernel functions, and RMSE is calculated for each. Table 4 shows the average localization accuracy of 100 different runs performed for each magnetic map.

From these experiments, we can see that both the Matern 3/2 and the Matern 5/2 kernels improved localization accuracy (average) compared to the most commonly used RBF kernel. With the Matern 3/2 having a slightly higher average but lower maximum errors, compared to the Matern 5/2. The standard deviations from both kernels are also considerably lower than that of RBF, showing that they are more stable.

The Exponential kernel showed the worst performance (both by itself and when combined with the cosine kernel). Therefore, it is considered that the Exponential kernel is not suitable for magnetic localization.

Interestingly, when the exponential kernel was combined with the Matern 3/2 kernel, while the average performance was worse than the Matern kernel by itself, it got the lowest maximum errors (1.86 m). This indicates that while the Exponential kernel does seem to yield lower localization accuracies, it does improve stability.

## 5. SGPR for Generating Large-Scale Magnetic Map

For autonomous movement using the magnetic navigation method, it is necessary to first obtain the magnetic intensity of the entire environment by sensing the magnetic intensity throughout the environment and then create a magnetic map by interpolating using GPR. In an environment such as the experimental environment shown in the previous Section 4, the number of magnetic intensities acquired does not need to be large, due to its size, so the magnetic map can be created by GPR without running out of memory. However, even in an indoor environment, for example, in a commercial facility such as a shopping mall, it is common that the environment is a large area, and the magnetic map cannot be created using GPR formulation as the computer would run out of memory. Therefore, we conduct magnetic mapping experiments using SGPR formulation described in Section 3.2.

As shown in Table 3, SGPR is a more efficient method than GPR, reducing the memory cost and computational complexity of computing the inverse of the covariance matrix.

SGPR prevents memory shortage for large datasets, which may occur in the GPR during magnetic mapping, enabling magnetic mapping of large areas.

In this section, we describe the experiment conducted for creating a magnetic map using SGPR and the robot localization experiment using the created magnetic map. We also verify whether the magnetic map using SGPR is effective for the autonomous movement using the magnetic navigation method.

### 5.1. Mapping

The robot used in the experiment and the method of acquiring the magnetic intensity is the same as in Section 4 (Section 4.2 for detailed explanation). In addition to the indoor environment of Section 4 (REAL), a new experimental environment in Haneda Innovation City (HICity) is employed. HICity is a commercial facility next to Haneda Airport in Tokyo, Japan. Magnetic maps are created using the magnetic intensities acquired in both locations.

The total number of acquired magnetic intensity data points in each environment was 5301 for REAL and 22,260 for HICity. The mapping is done using three different methods:GPR.GPR with clustered data (KM-GPR).SGPR.

In KM-GPR, k-means was used to create subsets of the data points; a map for each subset was then learned using an independent GPR). For all cases, the amount of memory consumption and the time required to create a map for each method are investigated and compared.

The kernel function used is the Matern 5/2 Kernel, which was found to be optimal for improving accuracy in Section 4. The inducing points are selected in consideration of the number of magnetic intensity data acquired in each environment as follows, and mapping experiments are conducted for each inducing point.

#### 5.1.1. Memory Consumption

We will verify the memory consumption during magnetic map creation in each experimental environment. The Gaussian process for magnetic mapping was programmed using the GPy library as in Section 4.2. The memory consumption is measured using the memory_profiler Python library (pythonprofilers/memory_profiler: Monitor Memory usage of Python code. https://github.com/pythonprofilers/memory_profiler (accessed on 1 March 2021)). The memory_profiler can measure the memory usage of a process as time-series data. In this verification, we measure the memory usage from the time the magnetic data required for training is loaded from a text file to the time the GPy library is used for training. The training data is the same for all three mapping methods. By using the above experimental method, we can compare the memory used in each creation method.

REAL, with 5301 magnetic data points, had the largest memory consumption (9.762 GiB) for mapping by KM-GPR (see Figure 7). GPR consumed the same amount of memory (4.049 GiB) as the first subset of KM-GPR, and it was able to create the map without any further increase. In SGPR, the memory consumption was the largest when the input data was read in for all patterns of inducing variable points, and only about 200 MiB was consumed during training.

In the case of HICity with 22,260 magnetic data points, the memory consumption was the largest (59.822 GiB) when GPR was used to create the map (see Figure 8). In the case of KM-GPR, which consumed the largest amount of memory in the magnetic mapping of REAL, it was possible to map without consuming a huge amount of memory (8.856 GiB) even though the input data was large. Even in the case of 2K inducing variable points, which is the largest number of inducing input points, SGPR was able to create maps with a smaller memory consumption (5.315 GiB) than GPR and KM-GPR. A server for big data, such as the NVIDIA DGX-1, can realize mapping with GPR.

As the robot is expected to be used in commercial facilities, our goal is to achieve stable operation through repeated autonomous movement experiments in real environments. This is required even in the presence of disturbances such as crowds, and we believe that it is necessary to conduct a series of actual experiments in the field. To conduct more experiments with the magnetic navigation method in a real environment, we need a system that can move autonomously immediately after mapping with a robot equipped with a PC that does not have high specifications. In such a case, magnetic map creation using SGPR is considered to be effective because due to its reduced memory consumption it can be used on portable computers.

Previous experiments have addressed memory consumption in the learning phase. For magnetic navigation, we also need to create a magnetic map. In our case, we create two grid maps per environment. One storing predicted mean magnetic intensity values and another storing predicted variance of the predictions for each grid of the free area of the grid map. Therefore, after the learning phase, we predict mean and variance values using the learned model, storing them in a text file. The memory consumption for creating such text files was also verified. The mapping environment and hardware used was DGX-1 with HICity. We compared GPR and the SGPR with 2K inducing points. Figure 9 shows the memory consumption for training and mapping the magnetic intensity distribution. Since GPR uses all the training and test data for prediction, the memory consumption during prediction as well as training is high, which caused memory errors on the program side, and we could not completely create the map. Unlike GPR, SGPR uses inducing points and test data, which reduces the amount of computation required to make a prediction. This allows the process from training to prediction to be done at reduced memory cost, with a peak memory consumption of 60.79 (GiB).We were able to achieve magnetic mapping without memory errors.

#### 5.1.2. Generating Time

Table 5 shows the results of the time taken to create the magnetic map. In the case of desktop PC, it took about 75 [s] for REAL in GPR, and in HICity, magnetic mapping could not be done due to forced termination of the program caused by insufficient memory. The proposed method can reduce the time required for REAL, and HICity can overcome the memory shortage and create a magnetic map except for the case of 2000 inducing points. Therefore, it can be said that the proposed method, SGPR, contributes to the reduction of memory cost and is effective for creating magnetic maps in large areas. Using DGX-1, REAL’s magnetic mapping with GPR took 127.6 (s), and HICity took 2315.4 (s). The DGX-1 is used to handle processes that consume huge amounts of memory rather than processing speed. Therefore, the processing time is larger than that of a desktop PC.

The above results are discussed. HICity has 22,260 magnetic data points, which is almost four times more than REAL. In the case of creating a magnetic map using the Gaussian process from more than 20,000 magnetic data points, the number of components in the inverse matrix calculation in GPR is $4.955\times 10^{8}$. When creating a magnetic map of an environment larger than HICity, the amount of magnetic information that can be obtained is likely to be even larger, forcing the calculation of an inverse matrix with an even larger number of components. This is not only limited to desktop PCs; even the NVIDIA DGX-1, which has considerable memory resources (512 GB), also had difficulties creating the desired magnetic maps. On the other hand, SGPR uses only the diagonal component for inverse matrix calculation, which is equal to the number of magnetic data, so the inverse matrix component required for magnetic mapping of HICity is only $2.226\times 10^{4}$. In summary, magnetic mapping using SGPR is a method that is more effective and contributes to reducing memory consumption when target environments are more vast and more magnetic information is acquired.

### 5.2. Effect to Localization

We verify the accuracy of the magnetic navigation method for localization using the magnetic map created by the proposed method. The experimental environment is REAL. We evaluate the accuracy of the localization method by comparing the localization results of the magnetic maps created by the GPR and the proposed method. The experimental results are shown in Figure 10. The results show that the navigation using the magnetic map created by the proposed method deviates from the path by about 1 [m] due to a decrease in estimation accuracy, as shown in Figure 10d. However, for other routes (Figure 10e,f), no significant deviation was observed as in the case of GPR (Figure 10b,c) and no degradation in estimation accuracy was observed. Therefore, it can be said that the magnetic map created by the proposed method is effective for the magnetic navigation method.

In localization in the magnetic navigation method, the likelihood calculation is performed using Equation (Equation 1). In this equation, $z_{map}$ and σ are the mean and variance, respectively, stored in each grid of the magnetic map calculated when predicting by the Gaussian process. The magnetic maps generated by GPR and SGPR, shown in Figure 10, show differences in the distribution of the magnetic intensity when viewed in a small area. However, the overall pattern of changes in magnetic intensity is similar, with large interpolations in areas of high magnetic intensity and small interpolations in areas of low magnetic intensity. Therefore, the magnetic navigation method, which uses the pattern of magnetic intensity disturbance as a landmark for localization, can be said to be capable of localization even when using a magnetic map based on SGPR, which is similar to the actual distribution of magnetic intensity. The variance stored in each grid is represented by a color map as shown in Figure 11. From this figure, we can see that the magnitude and distribution of the stored variance are different between GPR and SGPR. However, as can be seen from the color shading, the value of the variance stored in the magnetic map by SGPR is about 0.1 at most, which is not extremely large. Thus, the magnetic map of SGPR, which captures the pattern of changes in magnetic intensity and has less difference in the magnitude of the variance value compared to GPR, is considered to be an effective map for localization by the magnetic navigation method.

## 6. Conclusions

In this paper, we introduced a navigation method for autonomous mobile robots that use magnetic information, aiming for robust localization in crowded environments. The magnetic information is obtained by measuring the intensity of the geomagnetic field, which is not affected by nonmagnetic objects such as humans and is stable over time. Localization is performed by using the disturbance of the intensity as a landmark. The Gaussian Process Regression framework is used to create a magnetic map that stores the magnetic intensity in the environment, which is necessary for the navigation method. This regression method is capable of interpolating data with Gaussian distributions and predicts the intensity distribution of magnetic data acquired as points and uses it as a map.

However, Gaussian Process Regression requires $O(n^{3})$ and $O(n^{2})$ in terms of computational and memory costs, respectively, so it is difficult to handle large amounts of data. As a method to reduce the cost of each of these, we use an approximate method of calculation using the inducing variable method called the Sparse Gaussian Process Regression. Magnetic mapping was performed using this method, and the mapping time and memory consumption were investigated in comparison with Gaussian Process Regression.

As a result, we found that the memory consumption can be significantly reduced, and a large number of inducing variable points were set up for highly accurate interpolation, i.e., more accurate magnetic mapping.

However, this is a case of applying the method to hardware such as the NVIDIA DGX-1, which handles processes that consume huge amounts of memory, and is capable of high-speed processing. We also conducted a localization experiment for the magnetic navigation method using a magnetic map created using a sparse Gaussian process. As a result, the magnetic map with a rather small number of inducing variable points (five) was quite coarse, but the map roughly captured the pattern of change in intensity. In conclusion, magnetic mapping by the sparse Gaussian process enables us to map a vast environment even on a desktop PC, and it is also useful for magnetic navigation methods.

As a future issue, although the mapping for two environments, an indoor environment and a large commercial facility with an open deck, was conducted in this paper, we believe that this method is effective for other environments as well. In addition, as shown in Figure 3, this experiment was conducted using a robot with a large wheel diameter that is less prone to errors such as slipping during autonomous movement. However, there are not many general autonomous mobile robots with such a large wheel diameter, and it is not clear whether the map can cover errors during navigation.

On mapping, we have selected kernel functions for Gaussian processes (GPR), but the kernel functions selected for GPR have been used for SGPR and the optimal kernel functions for SGPR have not been selected. Although unrelated to magnetic mapping, several previous studies have explored the selection of kernel function to improve GPR and SGPR predictions [54,55]. Therefore, the selection of the kernel function in SGPR to improve the localization accuracy in the magnetic navigation method should be considered in the future.

It is also important to note that regardless of the models used for learning the map, autonomous navigation may fail if magnetic objects that do not exist during magnetic mapping are present during autonomous navigation. The magnetic navigation method is based on matching the magnetic map created in advance with the magnetic intensity observed during navigation. Therefore, if the measured magnetic intensity in navigation differs greatly from the magnetic map, localization may fail.

In the case of relatively small magnetic bodies such as smartphones, the effect on localization is considered to be small because the magnetic properties are not so strong as to significantly change the magnetic intensity in the environment.

However, automobiles and robots with large bodies have ferromagnetic materials such as engines and motors, which may have a large impact on the magnetic intensity in the environment.

To address this issue, previous research detected the presence of automobiles by measuring the geomagnetic field fluctuation caused by the automobiles [56,57]. In these studies, the intensity of the geomagnetic field fluctuating near the automobile was observed to be around 8 μG for a distance between the magnetic sensor and the automobile of 50 m and around 1 μG for a distance of 100 m.

By considering the amount of variation different types of objects can cause at different distances, it is possible to determine the possible affected areas, to avoid them. For example, areas where automobiles are expected to be located, such as parking lots, should be avoided.

In the case of narrow roads or environments where the robot cannot make such an action plan, it is recommended to temporarily switch to localization without using magnetic intensity, using only odometry, and then to move autonomously using the magnetic navigation method again after breaking through the magnetic fluctuation region. After returning to the magnetic navigation method, the errors that occurred during the temporary autonomous movement using odometry can be corrected, and stable autonomous movement is considered to be possible.

## Figures and Tables

**Figure 1 sensors-21-03972-f001:**
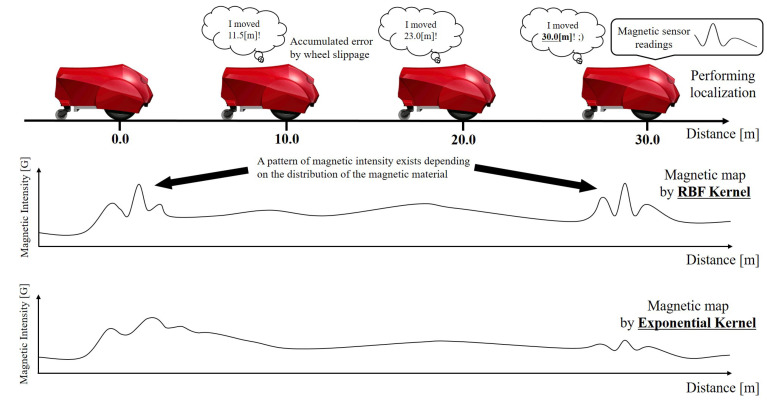
Travel distance estimation based on a magnetic fluctuation. The robot is given a magnetic map that stores the magnetic intensity with respect to distance.

**Figure 2 sensors-21-03972-f002:**
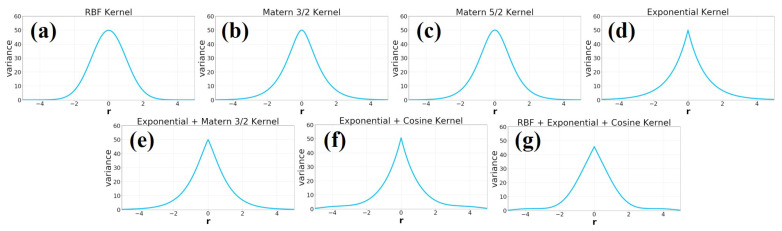
Differences in variance among the seven types of kernel functions proposed this time. r is the Euclidean distance of the input, $r\;=\;|x_{p}-x_{q}|$. (**a**) RBF Kernel (**b**) Exponential Kernel (**c**) Matern 3/2 Kernel (**d**) Matern 5/2 Kernel (**e**) Exponential + Cosine Kernel (**f**) Exponential + Matern 3/2 Kernel (**g**) RBF + Exponential + Cosine Kernel.

**Figure 3 sensors-21-03972-f003:**
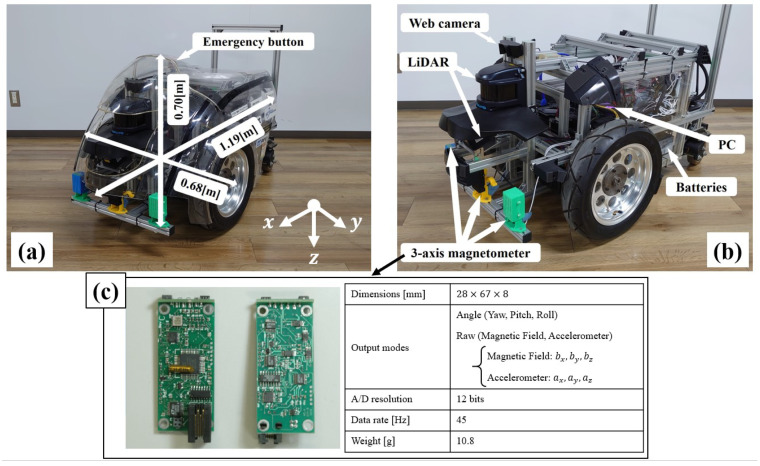
“MAUV”, the hardware used in the experiment. (**a**) MAUV with the cowl. The cowl was developed considering the requirements of a service robot for transporting people while ensuring safety [51]. (**b**) MAUV without the cowl. (**c**) Appearance and specifications of the magnetic sensors “3DM-DH”.

**Figure 4 sensors-21-03972-f004:**
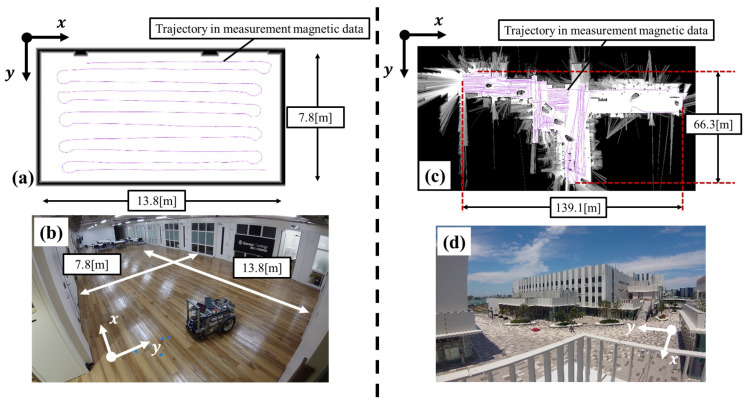
“REAL” and “HICity”, environments used in this experiment. (**a**,**c**) show the geometric map and the trajectory of the robot; when it measures magnetic information it is superimposed with a purple line. (**b**,**d**) show the environment in this experiment.

**Figure 5 sensors-21-03972-f005:**
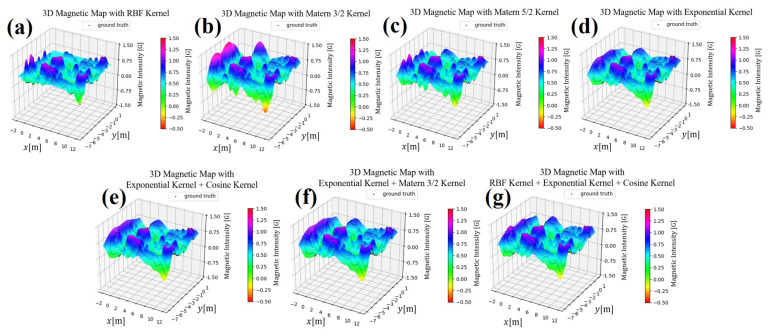
Magnetic map in the experimental environment. The strength is indicated by the color bar on the RGB scale. (**a**) RBF Kernel (**b**) Exponential Kernel (**c**) Matern 3/2 Kernel (**d**) Matern 5/2 Kernel (**e**) Exponential + Cosine Kernel (**f**) Exponential + Matern 3/2 Kernel (**g**) RBF + Exponential + Cosine Kernel.

**Figure 6 sensors-21-03972-f006:**
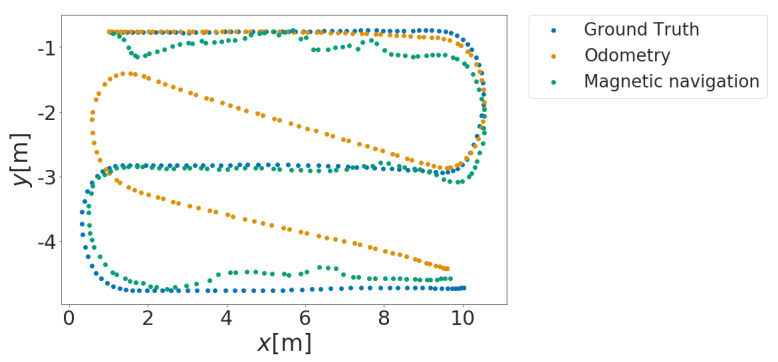
Example of localization result. Compare the results of odometry-only localization and magnetic navigation method localization for the specified path indicated by the blue dots with those indicated by the yellow and green dots.

**Figure 7 sensors-21-03972-f007:**
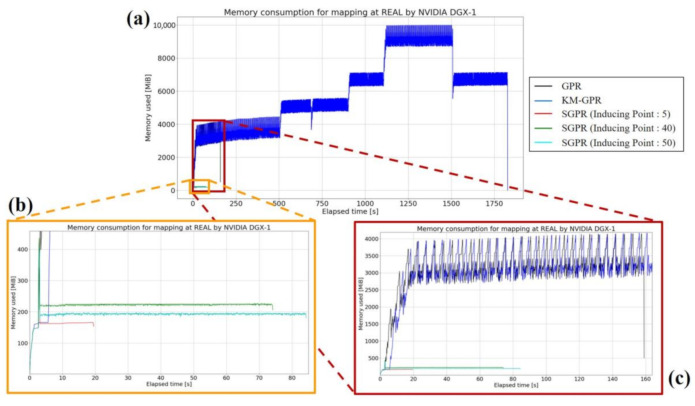
Graphs of memory consumption for each method in REAL’s magnetic mapping for GPR, KM-GPR, and SGPR with 5, 40, and 50 inducing points are shown. (**a**) Graph showing all methods, (**b**) enlarged graph of SGPR, and (**c**) enlarged graph of GPR.

**Figure 8 sensors-21-03972-f008:**
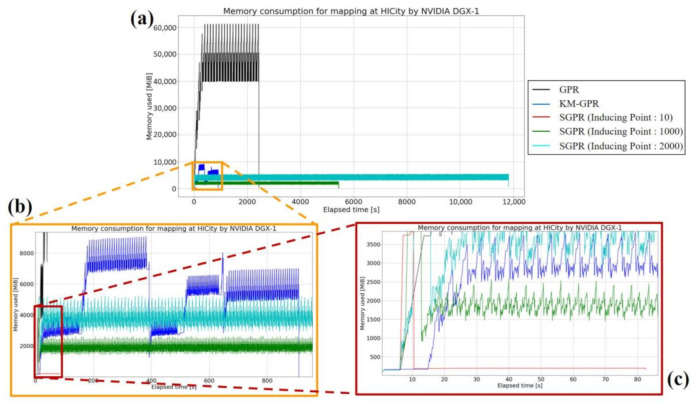
Graph of memory consumption for each method of magnetic mapping in HICity. GPR, KM-GPR, and SGPR with 10, 1K, and 2K inducing points are shown. (**a**) Graph showing all the methods, (**b**) enlarged graph of KM-GPR, and (**c**) enlarged graph of SGPR with 10 inducing points.

**Figure 9 sensors-21-03972-f009:**
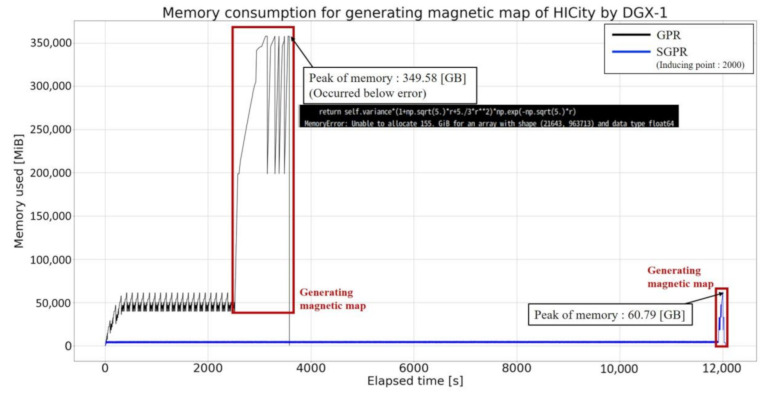
Graph of memory consumption for magnetic mapping of HICity using DGX-1.

**Figure 10 sensors-21-03972-f010:**
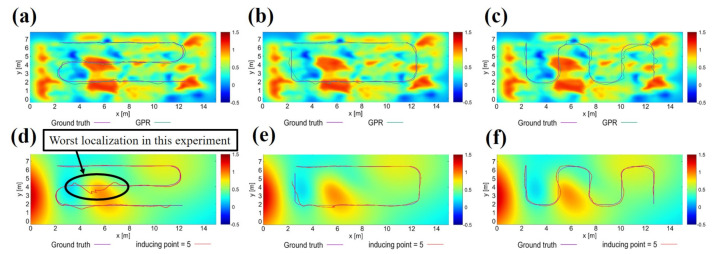
Experimental results of localization. Magnetic map created by GPR and SGPR is displayed as a two-dimensional color map, and each specified route and the trajectory of localization are superimposed. (**a**–**c**) and (**d**–**f**) show the results of localization for each specified route using the maps created by GPR and SGPR, respectively.

**Figure 11 sensors-21-03972-f011:**
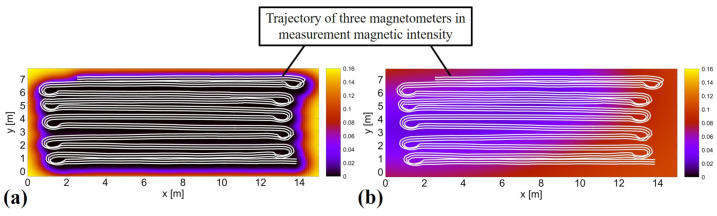
Color map of the variance stored in REAL’s magnetic map. (**a**) using GPR, (**b**) using SGPR with five inducing variable points. The white lines represent the trajectories of the three magnetic sensors mounted on the robot during the acquisition of magnetic information.

**Table 1 sensors-21-03972-t001:** Learning time and memory consumption when using the Gaussian process for publicly available magnetic data [41]. The hardware used in this case is shown in Table 2.

Path	# Sample	Elapsed Time (*mean* ± *2std*) [s]				Memory Consumption [MiB]			
		Full GP		SGPR		Full GP		SGPR	
		Desktop PC	DGX-1	Desktop PC	DGX-1	Desktop PC	DGX-1	Desktop PC	DGX-1
1	540	0.26 (±0.29)	0.68 (±0.78)	1.71 (±0.71)	2.11 (±0.79)	178.379	215.223	124.738	160.602
2	356	0.17 (±0.15)	0.45 (±0.45)	1.49 (±0.36)	2.21 (±1.65)	146.918	182.457	125.832	162.594
3	876	0.66 (±0.64)	1.55 (±1.66)	2.15 (±1.14)	2.52 (±0.77)	247.039	294.352	124.336	161.336
4	859	0.65 (±0.64)	1.59 (±1.56)	1.81 (±1.45)	2.36 (±1.14)	243.086	290.250	124.945	161.465
5	362	0.15 (±0.15)	0.38 (±0.39)	1.59 (±0.96)	2.02 (±0.97)	143.902	183.395	126.637	162.480
6	224	0.08 (±0.05)	0.22 (±0.14)	1.27 (±0.23)	1.88 (±1.22)	130.973	169.520	125.418	161.199
7	211	0.09 (±0.09)	0.26 (±0.26)	1.42 (±0.41)	1.69 (±0.44)	130.016	169.371	124.629	161.844
8	246	0.12 (±0.02)	0.24 (±0.07)	4.97 (±3.16)	2.55 (±1.27)	132.621	170.117	124.910	162.012
9	196	0.08 (±0.07)	0.23 (±0.15)	1.71 (±0.56)	1.94 (±0.44)	129.906	168.516	125.336	161.133
10	223	0.08 (±0.04)	0.22 (±0.12)	5.89 (±6.95)	7.81 (±4.16)	131.219	168.621	125.074	161.285
11	287	0.10 (±0.06)	0.26 (±0.17)	3.78 (±3.47)	6.08 (±5.72)	139.516	174.492	125.156	161.816

**Table 2 sensors-21-03972-t002:** Specifications of computers used to calculate required time and memory usage.

	Desktop PC	NVIDIA DGX-1 ^1^
Processor	Intel Core i7-9700 CPU@3.00 GHz × 8 ^2^	Dual 20-Core Intel Xeon E5-2698 v4 2.2 GHz
RAM	32 GB (Crucial CT16G4SFD8266 × 2 ^3^)	512 GB 2,133 MHz DDR4 RDIMM
Storage	Samusung SSD 860 EVO MZ-76E500 ^4^	4X 1.92 TB SSD RAID 0
OS	Ubuntu 20.04.1 LTS	Ubuntu 18.04.1 LTS

^1^https://www.nvidia.com/en-us/data-center/dgx-1/?ncid=van-dgx-1 (accessed on 1 March 2021); ^2^
https://www.crucial.com/memory/ddr4/ct2k16g4dfd8266 (accessed on 1 March 2021); ^3^
https://ark.intel.com/content/www/us/en/ark/products/191792/intel-core-i7-9700-processor-12m-cache-up-to-4-70-ghz.html (accessed on 1 March 2021); ^4^
https://s3.ap-northeast-2.amazonaws.com/global.semi.static/Samsung_SSD_860_EVO_Data_Sheet_Rev1.pdf (accessed on 1 March 2021).

**Table 3 sensors-21-03972-t003:** Memory and computational cost for each computational process in GPR and SGPR. *N* is input data points, *D* is input dimensions, and *M* is inducing points.

	Calculation Process	Memory Consumption Order	Computational Order
GPR	Covariance vector: $k_{*}$	O(ND)	O(ND)
	Covariance matrix: K	$O(N^{2}D)$	$O(N^{2}D)$
	Inverse matrix of K: $K^{-1}$	$O(N^{2})$	$O(N^{3})$
	Matrix product: $k_{*}^{T}K^{-1}m$	O(N)	$O(N^{2})$
SGPR	Covariance matrix: $K_{MM},K_{MN}$	$O(NM+M^{2})$	$O(NM+M^{2})$
	Diagonal matrix: Λ	O(N)	O(N)
	Matrix: $Q_{MM}$	$O(M^{2})$	$O(NM^{2})$
	Inverse matrix: $Q_{u}^{-1}$	$O(M^{2})$	$O(M^{3})$

**Table 4 sensors-21-03972-t004:** Results of localization experiments. The results of this experiment summarize the calculation of the mean and standard deviation as well as the minimum and maximum values from 100 localization experiments.

	RBF	Exponential	Matern 3/2	Matern 5/2	Exponential + Cosine	Exponential + Matern 3/2	Exponential + Cosine + RBF
Ave (±*2std*) [m]	0.616 (±1.01)	0.715 (±0.869)	0.578 (±0.638)	0.495 (±0.558)	0.788 (±1.46)	0.693 (±0.667)	0.619 (±0.654)
Min [m]	0.295	0.340	0.301	0.266	0.286	0.317	0.293
Max [m]	3.07	3.18	1.96	2.13	4.09	1.86	2.32

**Table 5 sensors-21-03972-t005:** Result of time when generating magnetic map (data:*mean*(±*2*×*std*)(s)). All the results are calculated as the mean and standard deviation of the 10 experimental results.As for the magnetic mapping in HICity, the time could not be measured in the case of GPR and 2K inducing variable points due to the forced termination of the program by insufficient memory.

**REAL**	Hardware	5	10	20	30	40	50	GPR
	Desktop PC	2.61 (±2.32)	4.55 (±1.28)	7.27 (±2.82)	10.9 (±0.72)	14.5 (±0.26)	17.7 (±0.52)	75.0 (±5.89)
	DGX-1	18.78 (±22.9)	29.30 (±17.34)	43.71 (±18.54)	58.26 (±0.98)	67.36 (±5.72)	77.77 (±1.72)	127.6 (±20.31)
**HICity**	Hardware	10	50	100	500	1K	2K	GPR
	Desktop PC	15.4 (±4.90)	78.7 (±0.66)	166.4 (±1.04)	1197.3 (±11.4)	2753.3 (±22.7)	-	-
	DGX-1	63.8 (±22.46)	228.1 (±2.65)	456.7 (±4.66)	2670.3 (±6.77)	5426.4 (±11.7)	11358.1 (±761.5)	2315.4 (±66.9)

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
