# Peer review of "Development of Magnetic-Based Navigation by Constructing Maps Using Machine Learning for Autonomous Mobile Robots in Real Environments"

_sensors, 2021, doi:10.3390/s21123972_

Round 1

Reviewer 1 Report

Major comments:

  • At the very minimum, wifi- and magnetic-based approaches should be mentioned in the introduction, and it should be clearly explained what is different in the proposed approach, and why it should be better than the state of the art..
  • For comparison magnetic-based approaches, either the authors should run a comparison using their dataset and by running other algorithms. Or run their algorithm on an existing dataset, such as "UJIIndoorLoc-Mag: A new database for magnetic field-based localization problems"
  • Special attention should be given to existing approaches that use GP for solving magnetic field, such as:
    • "Gaussian processes for magnetic map-based localization in large-scale indoor environments" DOI: 10.1109/IROS.2015.7354010
    • "Modeling magnetic fields using Gaussian processes"  
      DOI: 10.1109/ICASSP.2013.6638313
    • ....
  • The article needs extensive proofreading to improve the English.
  • It was not clear what in section 3 is specific to this paper, it looks like a general formulation of GPR, this could probably be summarized in a few equations needed to understand section 4/5 and adequate reference to GPR.
  • The memory consumption for the map creation is really big. What is the memory consumption when running?
  • How often should the map be recreated? The magnetic field fluctuates over time, how does that affect the accuracy of the localisation?

Minor comments:

  • Not sure why the coronavirus had to be mentioned, but I am pretty sure that it raged in 2020, not 2019.
  • l22-23 -> briefly mention the applications
  • l219 -> I think we can expect the reader to know that "In general, when a matrix has many zero elements, it is called a sparse matrix."
  • Please check the citation/references are properly generated before submission, l384 has a ?? for a reference to a section.

Author Response

Dear Reviewer 1,

thank you very much for your comments on our paper, from the grammar to the content and the experiments.

Please refer to the attached PDF file for my response to your comments. Please check it.

Please refer to pages 3 to 12 for our response to reviewer 1.

Once again, thank you very much for your review.

Reviewer 2 Report

This document presents a magnetic-based localization scheme by training a Gaussian Process Model (GPM). An approach to improve the computational performance is developed consisting of testing several kernel functions and introducing to the Gaussian Process Regression, the inducing variable method. The article is clear and well structured.

However, several issues should be addressed prior to publication:

  1. Authors claim that “no method has been used to change the kernel function as an approach to improve the localization accuracy”. Although not in the context of localization, the change of kernel function in GPM has been proposed in several articles, which should be included in the state-of-the-art revision, such as:

[a]. Teng, Tong, et al. "Scalable variational Bayesian kernel selection for sparse Gaussian process regression." Proceedings of the AAAI Conference on Artificial Intelligence. Vol. 34. No. 04. 2020.

[b]. Richardson, Robert R., Michael A. Osborne, and David A. Howey. "Gaussian process regression for forecasting battery state of health." Journal of Power Sources 357 (2017): 209-219.

  1. The above issue makes clear that the main paper contribution is not in the proposal of new methodologies or algorithms in the context of GPM, but in the application to magnetic-based localization in a large-scale and crowded environment. Unfortunately, the paper does not show any result on localization in a crowded environment. Perhaps the current outbreak does not allow such an experiment, but dynamic obstacles could be used instead of people.

  1. Several acronyms are used without a prior definition, such as SLAM, iid, FITC, etc.

  1. There are several typos, and a thoughtful revision should be performed. Some of them are as follows:
  2. Page 1, line 34, a space lack before (GPS).
  3. Page 3, item 3, reads “effctive”, should read “effective”.
  4. Page 3, line 112, a space lack before (GPR).
  5. In Table 1, entry (1,2) reads “calcuration” should read “calculation”.
  6. Page 14, line 384, there is a sign “??”, which should not be there.

  1. In both, figures and tables, captions are very extensive, which besides considering just a description, also an explanation is provided. Captions should be shortened in order to ease the reading of the document.

  1. On page 13, line 350 reads “This indicates that, while the Exponential kernel does seem to yield lower localization accuracies, it does improve robustness”. How should be understood “the robustness” in this context? The norm 1, was smaller for exponential + mater 3/2 kernel function was the smallest, but how it makes localization “robust”?

Author Response

Dear Reviewer 2,

thank you very much for your comments on the grammar, content, and discussion of our paper.

Please find our response to your comments in the attached PDF file.

Please refer to pages 13-19 for our response to Reviewer 2.

Once again, thank you very much for your review of our paper.

Reviewer 3 Report

This paper proposed a magnetic-based localization method. The experiment verifies its effectiveness.  It is suggested to correct some grammar errors and include recently published papers related to robotics localization before being accepted. 

Author Response

Dear Reviewer 3,

thank you very much for your comments on the grammar and content of our paper.

Please find our response to your comments in the attached PDF file.

Please refer to page 20 for our response to reviewer 3.

Once again, thank you very much for your review of our paper.
